# Updates on Group B *Streptococcus* Infection in the Field of Obstetrics and Gynecology

**DOI:** 10.3390/microorganisms10122398

**Published:** 2022-12-02

**Authors:** Yeseul Choi, Hyung-Soo Han, Gun Oh Chong, Tan Minh Le, Hong Duc Thi Nguyen, Olive EM Lee, Donghyeon Lee, Won Joon Seong, Incheol Seo, Hyun-Hwa Cha

**Affiliations:** 1Graduate Program, Department of Biomedical Science, School of Medicine, Kyungpook National University, Daegu 41944, Republic of Korea; 2BK21 Four Program, School of Medicine, Kyungpook National University, Daegu 41944, Republic of Korea; 3Department of Physiology, School of Medicine, Kyungpook National University, Daegu 41944, Republic of Korea; 4Clinical Omics Institute, Kyungpook National University, Daegu 41405, Republic of Korea; 5Department of Obstetrics and Gynecology, Kyungpook National University Chilgok Hospital, Daegu 41404, Republic of Korea; 6Department of Immunology, School of Medicine, Kyungpook National University, Daegu 41944, Republic of Korea

**Keywords:** Group B *Streptococcus*, obstetrics, gynecology, early-onset disease, late-onset disease, vaginosis, detection, intrapartum antibiotic prophylaxis, GBS vaccine, microbial therapy

## Abstract

Group B *Streptococcus* (GBS, *Streptococcus agalactiae*) is a Gram-positive bacterium that is commonly found in the gastrointestinal and urogenital tracts. However, its colonization during pregnancy is an important cause of maternal and neonatal morbidity and mortality worldwide. Herein, we specifically looked at GBS in relation to the field of Obstetrics (OB) along with the field of Gynecology (GY). In this review, based on the clinical significance of GBS in the field of OBGY, topics of how GBS is being detected, treated, and should be prevented are addressed.

## 1. What Is Group B *Streptococcus* (GBS)?

### 1.1. Microbiology

Group B *Streptococcus* (GBS, *Streptococcus agalactiae*) is a Gram-positive, β-hemolytic facultative anaerobic bacterium, comprised of cocci arranged in chains, that primarily colonizes the gastrointestinal and urogenital tracts [1,2]. In 1933, Lancefield identified different species of *Streptococci* based on their serological properties and hemolytic patterns; among these different species, *S. agalactiae* was found to belong to Group B [3]. Later, Group B was further divided into 10 serotypes (Ia, Ib, II, III, IV, V, VI, VII, VIII, and IX) based on the composition of the capsular polysaccharides (CPS) [4,5].

### 1.2. Virulence Factors

GBS has several virulence factors that facilitate its adherence to the host cell, evasion of the host immune system, colonization, and eventual progression to invasive GBS disease [6]. First, GBS has pili that play a role in its attachment to the host cell and further invasion into the cell [7]. In addition, GBS produces β-hemolysin, a pore-forming toxin that destroys the red blood cells of the host and causes hemolysis. Moreover, enzymes produced by GBS, such as C5a-ase, assist the bacterium in the evasion of the human immune system and further evolution to GBS infection. Lastly, the polysaccharide layer that encapsulates GBS is rich in sialic acid, which is also found in human cells. Therefore, the naïve immune cells of a newborn may recognize the sialic acid of GBS as that of human cells and allow the bacterium to survive in the body, leading to infection [8].

### 1.3. Epidemiology

In the 1970s, GBS emerged as a predominant pathogen causing sepsis or meningitis in newborns in the US and worldwide [9,10,11,12]. Since then, it has been investigated widely, especially in relation to invasive GBS disease in neonates. According to the US Centers for Disease Control and Prevention (CDC), approximately one in every four pregnant women carries GBS [13]. Although most of them are asymptomatic GBS carriers, its colonization in the maternal urogenital tract at the time of delivery is an important risk factor for neonatal GBS infection. In a recent report, approximately 19.7 million pregnant women (posterior median; an updated predictive median value after taking consideration of currently available data) were estimated to have rectovaginal GBS colonization in 2020, resulting in an estimated 58,300 infant deaths and 46,200 stillbirths [14]. In addition to perinatal or infantile death, invasive GBS infection in neonates was associated with long-term neurodevelopment impairments [14]. In terms of GBS incidence, there is a variance in GBS colonizer estimates by regions around the world. Overall, 18% of the world is estimated to be colonized, ranging from a high incidence in the Caribbean of 35% to a much lower prevalence in Southern Asia (13%) and Eastern Asia (11%) [15]. Among regions including North America, Europe, and Latin America, Sub-Saharan Africa had the highest burden of invasive GBS infection with 20,300, 90,800, 78,100, and 50,600 cases of stillbirth, early-onset disease (EOD), late-onset disease (LOD), and infant death, respectively, accounting for nearly half of all global GBS-related events [14].

## 2. GBS-Related Clinical Diseases

### 2.1. GBS and Non-Pregnant Women

The incidence of GBS disease in non-pregnant women or immunocompromised adults is increasing, especially in elders with underlying diseases [16]. In total, 20 to 70% of these infections are nosocomial [17]. Several clinical diseases result from GBS infection in non-pregnant adults. The first such disease is skin and soft tissue infection, which is the most frequently reported clinical manifestation associated with invasive GBS disease. This infection mostly presents as cellulitis, decubitus ulcers, and infected foot ulcers [16]. GBS pneumonia usually occurs in older adults with neurological impairments, such as dementia and cerebrovascular disease [16]. Approximately 5 to 23% of non-pregnant women with GBS infection have urinary tract infections, and most such infections occur in elderly adults [4]. Meningitis is another significant but not uncommon clinical manifestation in adults. Bone and joint infections, such as osteomyelitis and septic arthritis, and recurrent invasive GBS infection are other clinical manifestations in non-pregnant women [16].

### 2.2. GBS in Pregnancy

Maternal and fetal GBS results range from asymptomatic colonization to sepsis. It causes maternal bacteriuria, pyelonephritis, postpartum mastitis, and endometritis [8,18,19,20,21]. Although heterogeneity was noted globally, GBS serotypes Ia, III, and VI accounted for the majority of cases of maternal systemic GBS disease [15,22,23,24]. It may also reach the amniotic fluid by overcoming the normally protective cervical barrier during pregnancy. These ascending GBS infections have been involved in preterm labor, prelabor rupture of membranes (PROM), chorioamnionitis, fetal infection, and stillbirth [25]. About 98% of colonized newborns show a good prognosis while 1–3% of colonized newborns have early-onset disease (EOD), which is defined as neonatal infection within 7 days after birth [26]. The main causes of EOD in neonates are vertical transmission from the mother and amniotic GBS infection [8,27]. In the US, more than 95% of cases of EOD are related to GBS serotypes Ia, Ib, II, III, IV, and V [28]. In neonates with EOD, sepsis occurs in 80 to 85%, and 10% of EOD cases show pneumonia [18,28]. Lastly, meningitis is found in about 5 to 10% of EOD [28].

However, meningitis is a common manifestation in late-onset disease (LOD), an infection beyond 6 days through 90 days after birth. Although the pathogenesis of LOD is less understood, it is believed to be acquired from vertical transmission, nosocomial sources, or horizontal transmission from household or community settings [29,30]. The six aforementioned serotypes responsible for most cases of EOD are also present in more than 97% of cases of LOD, and among them, serotype III is known to be highly related to meningitis [28]. Although the risk factors of LOD are not yet well-established, the expected risk factors are maternal GBS colonization, prematurity, young maternal age, HIV exposure, and black maternal ethnicity [30]. Among them, prematurity was identified as a major risk factor in a recent study [31]. The clinical manifestations of LOD other than meningitis are bacteremia without a focus, bone-joint infection, and cellulitis-adenitis. Unlike EOD, the evaluation and initiation of empiric treatment for LOD are mainly based on the clinical appearance and signs of illness [32]. At present, there are no effective preventive measures for LOD.

Very late onset GBS disease (VLOD) is identified as GBS infection in infants aged 3 months or older. The risk factors of VLOD are believed to be similar to those of LOD [33]. However, most cases of VLOD occur in preterm babies or those with very low birth weight. Infants with VLOD are more likely to acquire immunodeficiency or HIV infection, and the most common clinical manifestations of VLOD are bacteremia without a focus and meningitis [8].

In infants, the mortality rate for EOD is about 2 to 3%, and that for LOD is 1 to 3% [8]. In preterm babies, the mortality rate for EOD is approximately 20 to 30%, and that of LOD is 5 to 8% [8]. Although GBS-infected infants survive and leave the hospital, their survival rate in the first decade is low, and they experience repetitive hospitalization in their first five years of life [18]. The findings of Yeo et al. support this high mortality rate in infants with EOD. Children with GBS infection are 3-fold more likely to die and require hospitalization during the first 11 years of life [34]. Long-term morbidity is another serious issue associated with GBS infection in infants. GBS infection can lead to high risks of permanent neurodevelopmental disorders, such as cerebral palsy and epilepsy [34]. For instance, only 51% of infants with GBS meningitis develop appropriately with age [35]. Meanwhile, 25% of such infants develop mild to moderate neurodevelopmental disorders, and the remaining infants experience severe neurological or functional impairments [35]. With its concerning and long-lasting comorbidities, it is critical to detect GBS infection at an early stage and prevent the manifestations of invasive GBS disease in neonates and infants.

In recognition of its clinical significance in maternal and neonatal health, the CDC and the American College of Obstetricians and Gynecologists (ACOG) have provided guidelines and recommendations for GBS screening and management in pregnant women focusing on the prevention of EOD (Figure 1). The ACOG recommends performing universal GBS screening between 36 0/7 and 37 6/7 weeks of gestation, which is approximately 5 weeks before their delivery [27,36]. If positive GBS culture results are obtained from the screening, intrapartum antibiotic prophylaxis (IAP) is indicated. They also commented that GBS bacteriuria at any concentration identified at any time in pregnancy represents the heavy maternal vaginal-rectal GBS colonization, which indicates the need for IAP without a subsequent GBS screening vaginal-rectal culture at 36 0/7–37 6/7 weeks of gestation. When bacteriuria with a concentration higher than 10^5^ CFU/mL is identified at any time during pregnancy, acute maternal antibiotic therapy should be administered during the antepartum period, and IAP is required at the time of birth [26]. If the concentration is below 10^5^ CFU/mL, maternal antibiotic therapy during the antepartum period is not required but IAP at the time of birth is warranted.

IAP is also administered during intrapartum screening, especially in pregnant women without GBS screening results who developed their labor or rupture of membranes before 36 weeks of gestation (Figure 1). When such women enter labor, IAP is continued until the birth of the baby. For women who have not entered labor, a different strategy is applied. In particular, IAP administration is discontinued, and further administration is decided based on the result of the universal vaginal-rectal GBS screening culture. IAP is administered at the onset of labor if the GBS culture result is positive. IAP is not administered to mothers with negative culture results, and another vaginal-rectal GBS culture is recommended when the 5-week screening accuracy window has passed. Lastly, when culture results are unavailable at the onset of labor, IAP is administered to mothers before 37 weeks of gestation. For those with more than 37 weeks of gestation, IAP is administered after considering the following risk factors: longer duration of ROM (>18 h), the occurrence of PROM, and maternal intrapartum temperature exceeding 38 °C [26]. The indications for IAP administration to prevent EOD recommended by the CDC are described in Table 1 below [26].

The detection of GBS colonization in the maternal rectovaginal area by universal GBS screening culture in pregnant women at 36 0/7–37 6/7 weeks, identification of candidates of IAP based on risk factors, and appropriate IAP administration are the mainstream procedures for the prevention of EOD [27]. These methods have substantially reduced the prevalence of neonatal EOD to approximately 0.25 cases per 100 live births, representing a nearly 85% decrease compared with that in 1990 [28,30,36,37,38,39,40]. However, LOD has not been prevented, and its rate remains stable, at approximately 0.27 per 1000 live births [30,41]. Acknowledging the significance of such guidelines for the prevention of GBS disease, many other countries and clinical institutions have also proposed guidelines for treating maternal GBS infections as well [42,43,44,45,46,47].

Penicillin G is usually the first-line agent for GBS IAP because of its low cost, low toxicity, and narrow-spectrum activity [2]. According to the American Academy of Pediatrics, antibiotics should be administered at least 4 h before delivery to achieve a sufficient concentration in the amniotic fluid and in the placenta circulation, which would reduce maternal-fetal GBS transmission [48]. However, for those who are allergic to penicillin G, or β-lactam antibiotics, other antibiotics are alternatively given based on the risk-based algorithm. If pregnant women are allergic to β-lactam antibiotics but have a low risk of anaphylaxis, cefazolin is given. For those with a high risk of anaphylaxis, a susceptibility test for clindamycin should first be performed via a GBS culture method. Clindamycin should be administered to pregnant women with clindamycin-susceptible GBS infection, and those with clindamycin-resistant GBS infection should be administered vancomycin. The treatment algorithm involving the usage of antibiotics is described in Figure 2 [49,50]. 

Unlike other general antibiotic therapies, IAP is only given as a “partial antibiotics treatment”. For example, “full antibiotics therapy” has been employed for treating gastric cancer to eradicate *Helicobacter pylori*, a bacterium that was classified as a group 1 human carcinogen by the International Agency for Research on Cancer of the WHO [51,52,53,54,55,56,57]. Such full doses of antibiotics would destroy *H. pylori* colonization and therefore treat gastric cancer. However, in the field of obstetrics, such “full antibiotics treatment” cannot be performed because it can cause severe harm to the health of both the mother and fetus, including fatal disease or chronic disabilities.

Similarly, IAP has some limitations. There has been an increasing necessity for antimicrobial susceptibility tests for pregnant women because of the risk of anaphylaxis [58,59]. Some researchers reported that the ratio of maternally transferred antibodies to newborns is approximately 0.5–0.7, indicating the inefficiency of IAP treatment [60]. As previously mentioned, IAP possibly disrupts the microbiota of newborns, and treatment in that neonatal period has inherent risks [1,2,61,62,63].

### 2.3. GBS and Maternal Microbiome

As the vaginal flora is one of the microbiomes in the maternal vaginal environment, many studies have investigated its association with GBS [64,65]. Moreover, many studies have examined the association of the vaginal microbiome with adverse obstetric outcomes in the context of vaginosis, an imbalance of the vaginal microbiome composition that usually involves the loss of *Lactobacillus* and the overgrowth of other pathogenic microbiomes [66,67,68,69,70,71]. There are mainly two types of vaginosis: bacterial vaginosis (BV) and aerobic vaginosis (AV). BV is defined as the replacement of normal lactobacilli by a large number of anaerobic microbes, such as *Gardnerella*, *Prevotella*, or *Bacteroides* [72]. AV is characterized by the disruption of lactobacilli accompanied by increases in the number of aerobic facultative pathogenic microbes, such as GBS or *Escherichia coli* [73]. Both BV and AV are considered to be associated with various serious obstetric clinical complications, such as preterm birth, miscarriage, prelabor rupture of membrane (PROM), fetal infection, and low birth weight [74]. Donders et al. reported that both BV and AV along with an abnormal vaginal microbiome in early pregnancy are associated with such complications [75,76]. The same group further confirmed an abnormal vaginal flora can influence cervical shortening, thereby leading to preterm birth [77].

Recently, Mohamed et al. discovered that dysbiosis in pregnant women with BV was accompanied by an increased prevalence of *Streptococcus*. In particular, pregnant BV showed a significant decline in the abundance of *Lactobacillus* (34.7% vs. 88.4% in the healthy group) along with an increase in the abundance of *Streptococcus* (29.7% in the BV group vs. 3.8% in the healthy group) [78]. However, Daskalakis et al. reported that although BV was associated with an increased risk of preterm delivery, GBS colonization in the second trimester of pregnancy was negatively correlated with the risk of preterm birth [79]. 

Hypothesizing that the presence of GBS in pregnant women influences and perhaps leads to obstetrics-related complications, such as PROM or preterm birth, we conducted a 16S rRNA metagenomics study using swab samples from nine pregnant Korean women as a pilot study. Contrary to our expectation, there was no significant change in alpha and beta diversity between the GBS-positive and GBS-negative groups. This finding was consistent with that of other studies on pregnant Korean and Guatemalan cohorts [80,81,82]. However, we identified four relatively abundant pathogens in the GBS-positive group: *Actinomyces*, *Shigella*, *Fenollaria*, and *Gemella.* We presume that these genera could serve as potential indicators of pregnancy management and stratification [unpublished data]. Our findings might further support the dynamics between GBS and other diverse vaginal microbiomes in the vaginal environment, which collectively influence pregnancy-related adverse outcomes.

## 3. GBS in Gynecology

Recently, the concept of vaginosis and its relationship with gynecological malignancies have been studied widely. Vaginosis refers to the disruption of the healthy balance in vaginal microbiome composition, which is normally predominated by *Lactobacillus* species that sustain the low pH of the vaginal environment and therefore block other pathogens [83,84,85,86]. Loss of protection from *Lactobacillus* and changes in the relative abundance of other microbes from such imbalances are believed to be associated with gynecological malignancies, such as endometrial cancer, ovarian cancer, cervical intraepithelial neoplasia (CIN), and cervical cancer (CC) [64,73,87,88]. However, only a few studies have investigated gynecological malignancies in relation to GBS specifically.

Dysbiosis in the cervico-vaginal microbiome in relation to the development of cervical pathology or a higher risk of human papillomavirus (HPV) infection has been asserted [89,90,91]. Some studies have reported that community state type (CST) IV, which is characterized by the lack of *Lactobacillus* dominance, is mainly associated with BV and AV [92]. Bacterial dysbiosis in combination with HPV infection can serve as a risk factor for CIN, and the progression of CIN was associated with increased vaginal microbiome diversity [93,94,95]. One study examined the cervical microbiota in relation to various stages of CC and found that CST IV, which is predominated by GBS (7%), is significantly related to the risk of squamous intraepithelial lesions (SILs). They further discussed the possibility that the cytokine profile modified as a result of changes in cervical microbiome composition can influence the cervical microenvironment during the development of SILs and CC [96]. Zhang et al. also investigated the cervical microbiome composition in relation to HPV infection and CIN severity and identified both direct and indirect associations between the microbiome and CIN status. GBS was only indirectly associated with CIN severity. They further argued that such direct and indirect effects of microbial composition would influence the risk of HPV infection and cervical carcinogenesis [97]. We also investigated microbiomes in relation to the progression of cervical cancer. To acquire different cervical cancer stages, we employed a machine-learning algorithm. The representative species were *Lactobacillus* for healthy controls; *Gardnerella* for patients with CIN, and *Streptococcus* as a potential biomarker discriminating invasive CC from CIN [98]. In addition, we found that GBS was associated with HPV infection and CIN2+ lesions [99]. However, a previous study reported that there was no correlation between HPV infection and GBS [100].

Few studies have investigated endometrial cancer in relation to the microbiome. Hakimjavadi et al. recently discovered that the vaginal microbiome segregates endometrial carcinoma from benign gynecological conditions and holds strong potential as a predictive marker of cancer grade and histology [101]. Another study found a significant difference in the endometrial microbiome at the genus level, and *Streptococcus* was one genus that was significantly more abundant in the healthy control group than that in patients with endometrial polyps and chronic endometritis [102,103]. The association of an imbalanced cervico-vaginal microbiome or oncobiome with ovarian cancer has been investigated in previous studies as well [104,105]. These studies further argued that such oncobiosis can lead to lower microbial diversity, thereby eventually transforming the immune system and causing pathogenesis [104,105]. Furthermore, Banerjee et al. identified microbial signatures uniquely associated with ovarian cancer, and *Streptococcus* was one of the bacterial genera detected in ovarian cancer [106]. Although this study did not find the exact association between malignancies and GBS, the approaches used therein provide future directions for examining the potential role of GBS in the vaginosis environment and its influence on and association with gynecological malignancies.

Considering the relatively recent trend in microbiome-related studies in the field of gynecology compared with obstetrics, fewer studies have investigated the interactions or roles of GBS in gynecological malignancies. However, many researchers have stressed the necessity of investigating the relationship between microbiome composition and gynecological malignancies, and GBS is one of them. Moreover, recent research supports the significance of GBS within vaginosis and its role as an indicator of malignancy propagation or carcinogenesis. The development of high-throughput sequencing approaches would further provide an in-depth understanding of the dynamics of GBS in the field of gynecology. Therefore, more solid and comprehensive findings are expected in near future.

## 4. Detection, Prevention, and Treatment of GBS

### 4.1. Detection: Various Detection Methods for GBS

For the aforementioned reasons, early detection of GBS is extremely critical. Various methods are currently being used for the detection of GBS, including conventional culture methods, enrichment culture methods, and molecular genetic methods, such as polymerase chain reaction (PCR). Among these, the current gold-standard GBS detection methods recommended by the CDC are selective enrichment culture methods [36].

In conventional culturing methods, the collected swabs are inoculated on blood agar plates and aerobically incubated at 37 °C for 48 h in a microbial incubator. Then, β-hemolytic colonies, which are the indicators of GBS, are inspected by specialists [107,108]. Although this method is the least expensive among the abovementioned three methods, it requires a longer time for GBS detection and holds a greater risk of generating false negative results due to underdetection caused by the overgrowth of other bacteria, maternal recolonization after performing cultivation, poor sampling techniques, or mishandling of the sample [1,109]. The CDC also stated that using direct agar plating instead of selective enrichment broth has the risk of producing extremely high false-negative results for GBS [110].

The second method is enrichment culture, which utilizes a GBS-selective medium and subsequent subculture on agar plates [108]. Several types of enrichment broths are used, including non-selective, selective, and differential broths [111]. Selective broths, such as Trans-Vag broth and Lim broth, inhibit and suppress the growth of enteric organisms and specifically enrich GBS. Differential enrichment broths, such as carrot broth and Granada biphasic broth, incorporate chromogenic pigments to detect only β-hemolytic strains of GBS [111,112,113]. Presumptive identification of the GBS enrichment culture can be made through the CAMP test [111,114] or serologic test with GBS antisera [115]. GBS produces a pore-forming toxin known as a CAMP factor which enhances the hemolysis of *S. aureus* [116]. 

Lastly, the nucleic acid amplification testing (NAAT) method utilizes PCR or loop-mediated isothermal amplification techniques. It specifically amplifies a section of the *cfb* gene in the GBS chromosome, which encodes a CAMP factor for the detection of GBS [117]. Various NAAT methods, such as BD MAX, GenomeEra CDX system, and Xpert GBS rapid test, are used for GBS detection [107,118,119,120,121,122,123]. BD MAX is an automated real-time PCR for GBS detection where a cervico-vaginal swab taken from pregnant women is specifically enriched in Lim broth and aerobically incubated for 18–24 h. Then, part of the broth is utilized for the assay [120].

NAAT-based detection holds many benefits. First, it can handle large numbers of samples with higher sensitivity and specificity. In addition, it provides the result much faster than culture methods. Such rapid diagnosis is beneficial when providing GBS results for pregnant women who did not undergo perinatal GBS testing but are at risk of preterm labor, PROM, or are approaching delivery [1,107]. However, its high cost limits its use in a wide range of the pregnant population, and it is not cost-effective to run only a few samples. In addition, its requirement of highly computational laboratory facilities makes it difficult to implement NAAT in resource-limited environments. Lastly, as it can amplify the dead debris of GBS nucleic acid, it carries a higher risk of giving false-positive results.

However, point of care testing (POCT) might overcome and resolve the abovementioned issues [124]. POCT provides results with much shorter turnaround times and cheaper costs with fewer professional requirements. It can also accelerate GBS screening before or in the absence of clinical visits. Early GBS detection would reduce the burden of OBGY professionals by reducing the number of risk factors to be considered as well as intervening in GBS progression into malignancy and thereby enabling the early prevention of GBS. Therefore, many inter-teams consisting of healthcare professionals, researchers, and hygiene educators are needed for better screening and prevention of GBS, thereby enriching communities and global populations.

### 4.2. Prevention: GBS Vaccination

To reduce global morbidity and mortality associated with GBS, developing a vaccine for GBS is critical [125,126,127,128,129]. Estimates suggest that if a GBS vaccine is adopted by 70% of pregnant women, almost 50,000 GBS-related deaths and more than 170,000 preterm births can be prevented annually [34]. However, no ‘licensed vaccine’ is currently available for GBS prevention [130].

After a consultation specifically concerning the development of vaccines for maternal immunization in 2016, the World Health Organization (WHO) declared an urgent need for a GBS vaccine to protect infant health and lives worldwide [131,132]. It also stated a strategic goal to develop safe, effective, and affordable vaccines for maternal immunization during pregnancy in order to prevent GBS-related stillbirth and invasive GBS disease in neonates and young infants [125].

At present, two types of GBS vaccines have reached phase 2 or 3 trials [130,133,134]. The first is a multivalent CPS conjugate vaccine designed to target the majority of disease-causing serotypes, and the other is a protein subunit vaccine [135,136]. Bianchi-Jassir et al. argued that a CPS-protein conjugate vaccine can target the majority of disease-causing serotypes, and therefore, it has the potential to prevent 95% of cases of maternal invasive GBS disease, 99% of stillbirths, and 99% of cases of neonatal GBS disease [137]. The other method, protein-based vaccines, has the potential to provide broader protection across all GBS serotypes and alleviate serotype replacement (i.e., capsular switching) [61,137].

Many pharmaceutical companies, such as Pfizer and MinervaX, have been trying to develop vaccines for GBS. Pfizer recently announced that its investigational GBS vaccine candidate, GBS6 (PF-06760805), which is designed to protect against the six most prominent GBS serotypes accounting for 98% of cases of GBS disease, received a designation of breakthrough therapy from the US Food and Drug Administration [138]. MinervaX’s GBS vaccine candidates in development are based on traditional CPS-conjugate technology, and the company is preparing for phase 3 trials [139,140].

There have been many clinical trials that evaluated these traditional GBS vaccines as phase 1 or 2 with human cohorts. For example, NCT03807245 investigated a recombinant protein GBS vaccine (GBS-NN/NN2) in different doses with a placebo; a total of 60 healthy female subjects aged from 18 to 40 were double-blinded and given 25 µg or 50 µg of GBS-NN/NN2 [141]. NCT02046148 examined the safety and immunogenicity of a trivalent GBS vaccine, a vaccine containing CPS from GBS serotypes Ia, Ib, and III and conjugated to the *Corynebacterium diphtheriae* CRM197 carrier protein, in healthy US pregnant women with a placebo as a phase 2 clinical trial [142]. The vaccine not only presented a favorable safety profile but also generated antibodies that transplacentally transferred to infants and persisted more than 3 months after the vaccination [143]. NCT01193920 also evaluated the safety and immunogenicity of a trivalent GBS vaccine as a dose-ranging study in both pregnant (between 28–35 weeks gestation) and non-pregnant women as a phase 1b trial study in South Africa; three different doses were given to pregnant women and one dose to healthy non-pregnant women [144]. A phase 1/2 study that was completed in July 2019, NCT03170609, evaluated three dose levels of a multivalent GBS6 vaccine with healthy adults from 18 to 40 with no history of previous GBS vaccination [145]. According to Absalon et al., all healthy adults tolerated the vaccine in all dose levels and elicited robust immune responses that persisted 6 months after the vaccination [146]. Lastly, NCT04596878 is an ongoing phase 2 clinical trial, which is expected to be completed by May 2021, that investigates the GBS-NN/NN2 vaccine in pregnant women with and without HIV [147]. Having experienced the necessity and importance of vaccines since COVID-19, much more diverse and developed GBS vaccines, such as mRNA vaccines, are expected in near future as well.

The deployment of affordably-priced vaccines would significantly reduce the burden of GBS in low- and middle-income countries where IAP administration is challenging. In addition, this strategy would potentially prevent the majority of cases of GBS-related disease without the adverse effects of IAP. Despite their merits, the limitations of GBS vaccines include a high cost, lack of coverage of all GBS strains, and the possibility of resistance. Therefore, some researchers argue that it is important to detect GBS before infection advances or progresses to a malignant state.

### 4.3. Another Possible Treatment Approach, Microbial Therapy

Current antibiotic administration is a satisfactory preventive method for GBS infection because of its low cost and high applicability, especially in low socioeconomic classes or resource-limited countries. However, the increased risk of anaphylaxis and the possibility of inherent severe neonatal risks remain its limitations. On that note, microbial therapy can be an alternative treatment approach.

Microbiome therapeutics became a hot topic in the field of obstetrics, gynecology, and translational research. Several reports and trials have examined gut microbiome treatment, such as fecal microbiome transplantation (FMT) for cancer therapy [148,149]. Additionally, some studies have reported that the composition of the gut microbiome modulates immune response mechanisms, such as anti-tumor activity, and thereby generates microbiome-tumor interactions [150]. Such microbiome-modulated mechanisms might be direct, but their specific downstream pathways remain to be elucidated [151]. For microbial therapy in general, known biomarkers are used as diagnostic tools to screen and monitor patients. After stratifications, microbial therapy is then applied [152].

Microbiome-based therapeutics are used for treating several diseases and are applied using diverse approaches, including dietary interventions, prebiotics, probiotics, synbiotics, postbiotics, phage therapy, and FMT. According to Gulliver et al., each approach holds both advantages and disadvantages. For example, probiotics are considered relatively safe. However, they do not target the disease specifically and only provide a temporary therapeutic response. In addition, the outcome of probiotic therapy depends on the specific microbiome colonization or gut microenvironment. Phage therapy is a highly-specific targeted method. However, it might require a specific environment for activation, and its effects might be limited to the disruption of the microbiota [152,153]. 

Such diverse microbiome-based therapeutics can also be applied to the vaginal microbiome as novel therapeutic approaches for GBS-infected patients. As dysbiosis of the vaginal microbiome is closely related to gynecologic malignancies and adverse obstetric outcomes as previously mentioned, manipulation of the vaginal microbiome has the potential to change clinical approaches in women [154,155]. With an emphasis on changing the vaginal microbiome, microbiome-based therapeutic modalities can be used similarly as described for the gut microbiome [156]. Both probiotics and prebiotics can be used to shift an unbalanced vaginal composition, mainly focusing on increasing *Lactobacillus* counts. Synbiotics, which combine probiotics and prebiotics, aim to overcome the limitations of prebiotics, i.e., the dependency on the presence of Lactobacilli. However, symbiotics might require a specific environment for their activity. For exploiting their characteristics as bacteria-specific viruses, phages can be utilized. Phages bind to specific receptors on the bacterial cell wall and insert their engineered therapeutic materials into the host cell, thereby eliciting promising effects [156].

Biofilm-disruptive agents represent another therapeutic option. Vaginosis refers to the high diversity of the microbial community dominated by pathogenic bacteria instead of *Lactobacillus*. Such polymicrobial infections generate biofilm on the vaginal epithelium and produce short-chain fatty acids which eventually increase the pH of the vaginal environment and later induce inflammation [157]. Wu et al. reported that treatment with antibiotics alone can reduce the microbial diversity and restore the *Lactobacillus* population, but they do not fully disrupt the biofilm. Therefore, treatment with antibiotics together with biofilm-disrupting adjuvants would represent a more comprehensive therapy.

Lastly, vaginal microbiome transplantation (VMT) is another microbial therapy method for treating vaginal dysbiosis [158]. In this treatment method, donors are recruited and medically assessed. Their microbiomes are screened for donation availability using microscopic evaluation or next-generation sequencing. Then, the optimal vaginal microbiota is transplanted to the recipients [159]. Lev-Sagie et al. reported that VMT remarkably alleviated the symptoms of patients and successfully restored the vaginal microbiome composition, including high *Lactobacillus* counts [159]. However, this relatively new approach remains controversial. Therefore, thorough and constructive regulatory standards for screening processes should be implemented to diminish the possible risk of transferring pathogenic microorganisms, particularly those that can cause antibiotic resistance [160].

As GBS may play a role in vaginosis and further lead to various adverse OBGY outcomes, targeting GBS specifically or targeting the overall microbiome composition to treat GBS-infected patients would be possible via such microbiome-based therapies. For the actual application of microbial therapy in GBS treatment, more in-depth studies are needed, and a better understanding of various dynamics, such as the host-microbiome and microbiome-microbiome interactions involving GBS, is required.

## 5. Closing and Future Directions

The presence of GBS implies serious clinical outcomes for young infants and neonates. However, for older individuals with GBS infection, the infection itself is less fatal. When considering that GBS might play role in the development of vaginosis, which leads to OBGY malignancies, there is a possibility that GBS is not a single primary causative but rather a critical factor that induces adverse outcomes. GBS might play a critical role in the development of serious clinical symptoms but is masked by other multifactorial components, making its detection difficult. GBS can even work as a strong hibernating pathogen that manipulates and modulates other bacteria, thereby leading to serious clinical outcomes.

Therefore, it is critical to investigate the mechanisms by which GBS interacts with and influences bacterial and host environments. More studies examining the etiology of GBS and its working mechanism using high-throughput sequencing techniques, such as RNA-seq, metagenomics, and metabolomics, are necessary. Moreover, inter-professional discussions and collaboration studies should be encouraged to establish better GBS management strategies aiming to control and reduce maternal-neonatal mortality and morbidity caused by GBS infection. Such integrative approaches would not only provide a better understanding of GBS but also benefit health at the national and global levels.

## Figures and Tables

**Figure 1 microorganisms-10-02398-f001:**
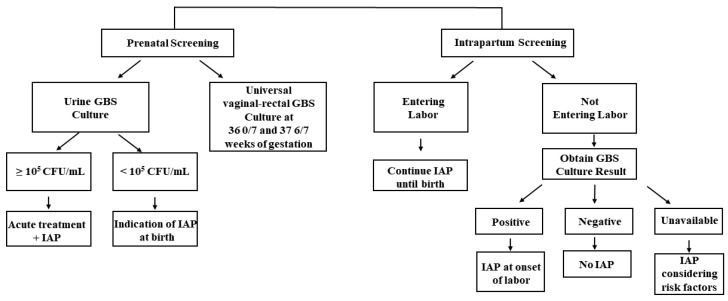
An overall schema of the CDC recommended IAP strategy for GBS screening and IAP administration [26].

**Figure 2 microorganisms-10-02398-f002:**
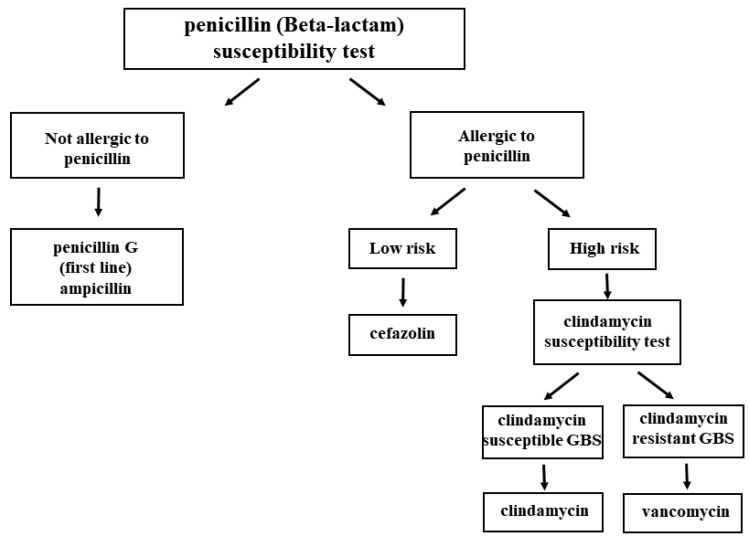
The algorithm recommended by the CDC for IAP treatment [26].

**Table 1 microorganisms-10-02398-t001:** Indications of IAP to prevent neonatal GBS EOD [26].

GBS IAP Indicated	GBS IAP Not Indicated
Maternal History Previous neonate with invasive GBS disease	Colonization with GBS during pregnancy (unless colonization status in current pregnancy is unknown at the onset of labor at term)
Current Pregnancy Positive GBS culture obtained at 36 0/7 weeks or more during current pregnancy (unless birth is performed before the onset of labor for a woman with intact amniotic membranes) GBS bacteriuria during any trimester of the current pregnancy	Negative vaginal-rectal GBS culture obtained at 36 0/7 weeks of gestation or more during the current pregnancy Cesarean birth is performed before the onset of labor on a woman with intact amniotic membranes, regardless of GBS colonization status or gestational age
Intrapartum Unknown GBS status at the onset of labor (culture not done or results unknown) and any of the following: ∘ Birth at less than 37 0/7 weeks of gestation ∘ Amniotic membrane rupture 18 h or more ∘ Intrapartum temperature 38 °C or higher ∘ Intrapartum NAAT result positive for GBS ∘ Intrapartum NAAT result negative but risk factors are present (i.e., less than 37 0/7 weeks of gestation, amniotic membrane rupture 18 h or more, or maternal temperature 38 °C or higher) ∘ Known GBS-positive status in a previous pregnancy	Negative vaginal-rectal GBS culture obtained at 36 0/7 weeks of gestation or more during the current pregnancy, regardless of intrapartum risk factors Unknown GBS status at the onset of labor, NAAT result negative, and no intrapartum risk factors present (i.e., less than 37 0/7 weeks of gestation, amniotic membrane rupture 18 h or more, or maternal temperature 38 °C or higher)

Abbreviations: GBS, Group B *Streptococcus*; NAAT, nucleic acid amplification test.

## Data Availability

Not applicable.

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
