# Peer review of "Updates on Group B Streptococcus Infection in the Field of Obstetrics and Gynecology"

_microorganisms, 2022, doi:10.3390/microorganisms10122398_

Round 1

Reviewer 1 Report

This review is well written and included different aspects of group B Streptococcus. I only have a few comments for this manuscript:

Line 239-250: The paragraph describes the results from the authors but is unpublished data. In the review, it might not be proper to include information that did not being certified by the peer review process. Also, Figure 1 on pages 7-8, should be removed. 

Line 257: GBS in gynecology. In the present form, this section describes the relationship between vaginal microbiome composition and gynecological malignancies. There are few studies (and also no solid evidence) that investigated gynecological malignancies in relation to GBS; therefore, this section seems not to be related to the topic of the review article. 

Line 20: These alternative treatment methods seem to be too premature, please modified the title of this section. 

Minor comments:

Line 36: Group B should be group B Streptococcus.

Line 56-57: What is “posterior median” here mean?

Line 396: Corynebacterium diphtheriae, not Cornebacterium diphtheriae, and should be italic.

Author Response

Dear Reviewer,

First of all, thank you very much for your comment.

For L239-250, we have removed the Figure 1 following your advice. It is yet unpublished data, however, we would like to point out that no significant difference in alpha or beta diversity between GBS-positive and GBS-negative groups is in accordance with other studies performed with pregnant women cohorts. The data is currently under review in other journal.

Line 257: Although the topic of GBS in gynecological malignancies is rather recent compared to that in obstetrics, it is one of the hot topics that are being researched in the field of gynecology. As this review is an “update” that focused on GBS in clinical field, especially in obstetrics and gynecology, we thought it would be valuable to mention how GBS is currently being investigated in both fields.

Line 20: We have revised the title to “Another possible treatment approach, Microbial therapy”.

About the minor comments, we have corrected typos and added definition of “posterior median”, which is an updated predictive value in Bayesian statistics that takes account of currently observed data.  

Again, we would like to expressive our deep gratitude for your time and constructive comments.

Reviewer 2 Report

This manuscript is a comprehensive review of the basic bacteriology of GBS, its pathogenicity, recent epidemiological information, clinical presentation in the perinatal period, methods of eradication, detection, prevention with vaccines, and alternative treatment with flora transplantation. Although Streptococcus pyogenes and Streptococcus pneumoniae tend to be the most common streptococci in the medical and comedical fields, it would be very valuable to have such a comprehensive review of GBS as well.

L63-.

It is difficult to read the connection between the number of cases and each region. Please rewrite the sentence so that it is clear which region has how many cases.

Table 1 is very difficult to read. Improve it by using left-aligned and center-aligned text, or bold and non-bold fonts. Basically, the text should be left-aligned, and item headings should be written in the center, for example. Abbreviations are also extremely difficult to read because they are integrated with the text; move them to Legend, etc.

L 319-.

There is no description of the CAMP test, which is a characteristic detection method for GBS. I am not a clinician and do not know much about the current situation, but I wonder if this test is no longer used. It may be an obsolete method in clinical practice for the reasons described after L322, but the CAMP test should still be mentioned as an important aspect of GBS. Considering that this manuscript is a comprehensive review of GBS, I think it is "update" only if it includes the methods that were used in the past and the current treatment of those methods.

Author Response

Dear Reviewer,

First of all, thank you very much for your comments.

Following your advice, we have rewrote L63 more clearly and revised Table 1. The headings of the Table 1 are now center-aligned with bold fonts and the texts are left-aligned with both bold and non-bold fonts. Also, we have placed * in front of Abbreviations for better visualization of legend of the Table 1.

About the CAMP test, it is one of the GBS Laboratory Identification (presumptive) methods recommended by American Society for Microbiology in 2021 [1]. With your comments, we have added more materials about CAMP test with references.

Again, we would like to expressive our deep gratitude for your time and constructive comments.